# Microbiome Composition and Microbial Community Structure in Mosquito Vectors *Aedes aegypti* and *Aedes albopictus* in Northeastern Thailand, a Dengue-Endemic Area

**DOI:** 10.3390/insects14020184

**Published:** 2023-02-13

**Authors:** Rutchanee Rodpai, Patcharaporn Boonroumkaew, Lakkhana Sadaow, Oranuch Sanpool, Penchom Janwan, Tongjit Thanchomnang, Pewpan M. Intapan, Wanchai Maleewong

**Affiliations:** 1Department of Parasitology, Faculty of Medicine, Khon Kaen University, Khon Kaen 40002, Thailand; 2Mekong Health Science Research Institute, Khon Kaen University, Khon Kaen 40002, Thailand; 3Department of Medical Technology, School of Allied Health Sciences, Walailak University, Nakhon Si Thammarat 80161, Thailand; 4Faculty of Medicine, Mahasarakham University, Maha Sarakham 44000, Thailand

**Keywords:** microbiota, *Aedes aegypti*, *Aedes albopictus*, northeastern Thailand, dengue vector, mosquito vector, microbial community, microbiome diversity

## Abstract

**Simple Summary:**

This study characterized the microbiota associated with *Aedes aegypti* and *Aedes albopictus* larvae and subsequently emerged adults relative to the microbiota in water from their breeding sites in Thailand, a dengue-endemic area. The adults were processed shortly after eclosion and had not fed. Thus, their bacterial community must have been carried through from the larval stage (transstadial transmission). However, there were substantial changes in the representation of many taxa between the larval and adult stages. More bacterial genera were associated with *Ae. aegypti* than with *Ae. albopictus*. *Wolbachia* was dominant and was present at a significantly higher frequency in the *Ae. albopictus* adult males. The genus *Blautia* was particularly abundant in *Ae. aegypti*. Abundance of the genus *Aquabacterium* decreased from larva to adult in *Ae. aegypti* and *Ae. albopictus*. Adult *Ae. aegypti* females had greater proportions of *Wolbachia*, *Blautia*, *Clostridioides*, and *Shinella* than did males. The microbial community of *Ae. albopictus* larvae was dominated by the genus *Serratia*, while the genus *Wolbachia* was very abundant in adults of both sexes. In addition to demonstrating transstadial transmission, our results provide important information about microbial dynamics across mosquito developmental stages.

**Abstract:**

Bacterial content in mosquito larvae and adults is altered by dynamic interactions during life and varies substantially in variety and composition depending on mosquito biology and ecology. This study aimed to identify the microbiota in *Aedes aegypti* and *Aedes albopictus* and in water from their breeding sites in northeastern Thailand, a dengue-endemic area. Bacterial diversity in field-collected aquatic larvae and subsequently emerged adults of both species from several locations were examined. The microbiota was characterized based on analysis of DNA sequences from the V3-V4 region of the 16S rRNA gene and exhibited changes during development, from the mosquito larval stage to the adult stage. *Aedes aegypti* contained a significantly higher number of bacterial genera than did *Ae. albopictus*, except for the genus *Wolbachia*, which was present at significantly higher frequencies in male *Ae. albopictus* (*p* < 0.05). Our findings also indicate likely transstadial transmission from larva to adult and give better understanding of the microbial diversity in these mosquitoes, informing future control programs against mosquito-borne diseases.

## 1. Introduction

Mosquitoes of the genus *Aedes*, especially *Aedes aegypti* and *Aedes albopictus*, are the primary vectors of a number of arboviruses, particularly, the dengue virus (DENV), which causes dengue fever (DF). There are an estimated 390 million DENV infections annually worldwide [1,2]. Thailand, a tropical country, is particularly at risk of DF [3]. Mosquito vector control is currently the primary method for DF reduction [4]. Manipulation of mosquito microbiomes has recently emerged as a promising subject for research into innovative, ecologically friendly vector-control strategies [5]. In the past decade, there has been increasing interest in mosquito microbiome research, leading to large amounts of data on different mosquito species from diverse geographical locations and their habitats [6,7,8,9]. Microbes play a crucial role in the biology of mosquitoes [9]. Mosquito larvae are aquatic and acquire most of their bacterial community members from the water in which they live [5,10]. Moreover, the bacterial community is altered by dynamic interactions during life and varies substantially in composition depending on mosquito biology and ecology [6,9]. However, there have been few studies in Thailand comparing the microbiota of *Ae. aegypti* and *Ae. albopictus* and of water from their larval habitats. In the present study, we examined the diversity of bacteria in field-collected aquatic larvae and subsequently emerged adults of both species from locations in northeastern Thailand. Here, we described for the first time the microbial composition, abundance, and variability in the two main arboviral vectors, *Ae. aegypti* and *Ae. albopictus*, in northeastern Thailand (a dengue-endemic area) based on analysis of DNA sequences from the V3-V4 region of the bacterial 16S rRNA gene. The results provide important information for other basic and advanced studies on mosquito biology, microbial community, and microbial dynamics across mosquito developmental stages.

## 2. Materials and Methods

### 2.1. Study Area and Mosquito Larvae Collection

This study was conducted at Mueang Khon Kaen district, Khon Kaen Province, in northeastern Thailand. This province is also a high-risk area for dengue in Thailand, with an incidence rate of 23.09 per 100,000 at-risk population from January to November 2022 [11]. Surveys of mosquito larvae were conducted at six sites (three for *Ae. aegypti* and three for *Ae. albopictus*) (Figure 1). Larvae and samples of their habitat water were collected simultaneously at all six sites from abandoned containers (i.e., plastic buckets, big earthen jars, plastic bowls). Sites where *Ae. aegypti* was collected were close to homes, whereas *Ae. albopictus* was generally found in forests or patches of shrubby vegetation; larvae of the two species did not co-occur.

This study was conducted according to the Guidelines for Animal Experimentation of the National Research Council of Thailand and approved by the Animal Ethics Committee of the Faculty of Medicine, Khon Kaen University, Thailand (AMEDKKU 012/2022).

### 2.2. Processing for Study Samples

All larvae (>50 larvae per collection site) and their habitat water were collected from the study sites. Twenty larvae were randomly picked with a Pasteur pipette and thoroughly washed several times with distilled water, then three times with sterile distilled water containing 0.1% diethylpyrocarbonate (DEPC) (Amresco, Solon, OH, USA). Finally, the larvae were placed in TRIzol reagent and morphologically identified to species under a microscope based on the structure of their comb scales [12]. The identified larvae were divided into two equal groups for RNA (10 larvae per sample) and DNA (10 larvae per sample) extraction. The remaining larvae (>30 larvae per sample) were reared in insect rearing cages in water from their habitat with no additional feeding. Newly emerged adults (24 h post-eclosion, no sugar or blood feeding) were collected. Adult mosquitoes were killed by cold shock, followed by sex and species separation based on key morphological characteristics [13,14]. Mosquito abdomens were separated from the head-thorax and pooled for each collection site and sex. After collection of the adult mosquitoes, about 400 mL of the habitat water in which they had spent their larval life was filtered using a 0.45 µm Millipore sterile filtering system (Merck Millipore Ltd.). In total, twenty-four samples were obtained from three *Ae. aegypti* collection sites and three *Ae. albopictus* collection sites for DNA extraction and 16S rRNA gene sequencing (Table 1).

### 2.3. RNA and DNA Sample Preparation

RNA extraction protocols were used for samples of larvae to identify DENV by qRT-PCR. Extraction of viral RNA from pooled larvae (*n* = 10) from each sample site (the larvae were collected from the same sample sites as shown in Figure 1) was performed using QIAamp Viral RNA Mini Kits (Qiagen, Hilden, Germany) according to the manufacturer’s instructions. Total RNA was DNase treated and DNA was removed using Ambion TURBO DNA-free Kits (Life Technologies, Carlsbad, CA, USA) following the manufacturer’s instructions. Concentration of extracted RNA was determined using NanoDrop spectrophotometers (Thermo Fisher Scientific, Waltham, MA, USA). The qRT-PCR was used to screen for dengue virus infection in mosquito larvae using a primer pair based on the conserved region of the genomic RNA of all four serotypes of DENV (forward primer D1: 5′-TCAATATGCTGAAACGCGCGAGAAACCG-3′, and reverse primer: 5′-TTGCACCAACAGTCAATGTCTTCAGGTTC-3′) [15]. The real-time PCR reactions were performed in a total volume of 20 µL with a Power SYBR Green RNA-to-CT 1 step kit (Applied Biosystems, Waltham, MA, USA). The reactions were amplified using the QuantStudio™ 6 Flex Real-Time PCR System (Applied Biosystems).

DNA from pooled larvae and adult abdomens was extracted using the NucleoSpin tissue kit (Macherey-Nagel GmbH & Co. KG, Duren, Germany) according to the manufacturer’s instructions. DNA extraction from larval habitat water samples was performed using a QIAamp Fast DNA stool mini kit (Qiagen). Briefly, a piece of membrane filter through which the water (400 mL) had been passed was extracted in optimized buffers in combination with an inhibitEX buffer (Qiagen) to remove PCR inhibitors, as recommended by the manufacturers. DNA concentration and purity were monitored using a NanoDrop Spectrophotometer (Thermo Fisher Scientific) and electrophoresis through 1% agarose gels.

### 2.4. Bacterial 16S rRNA Gene Amplification and Sequencing

The next-generation sequencing analysis was based on the 16S rRNA gene of bacteria. The V3-V4 regions of the gene were amplified using universal region-specific primers 341F (5′-CCT AYG GGR BGC ASC AG-3′) and 806R (5′-GGA CTA CNN GGG TAT CTA AT-3′) (Novogene, Singapore) tagged with sample-identifying barcodes. PCR products of the proper size were selected by 2% agarose gel electrophoresis. The DNA fragments were end-repaired and A-tailed then further ligated with Illumina adapters. The libraries were generated and sequenced on a paired-end Illumina platform (Novogene). The sequences have been deposited to NCBI under the accession number PRJNA919511.

### 2.5. Bioinformatic Analysis

Paired-end reads were assigned to samples based on their unique barcodes and were then truncated by removing the barcode and primer sequences. Paired-end reads were merged using FLASH (V1.2.7) [16], which generated the raw reads. Quality filtering of these was performed under specific filtering conditions to obtain high-quality clean reads [17] according to the QIIME (V1.7.0) quality-control process [18]. The reads were compared with the SILVA138 database using the UCHIME algorithm to detect chimeric sequences, which were removed [19]. The end product of this process was a suite of effective reads.

From all effective reads, sequences sharing ≥97% similarity were assigned to the same operational taxonomic unit (OTU) by Uparse software (V7.0.1090) [20]. A representative sequence for each OTU was compared to the SILVA138 SSU rRNA database using the QIIME in Mothur method for species annotation at each taxonomic level (kingdom, phylum, class, order, family, genus, species) [21]. The OTUs in the individual samples were divided among eight categories (species, life-stage and gender, and habitat water), and the mean abundances of each OTU were computed using QIIME for further analysis. The relative abundance of OTUs was normalized using a sequence number standard corresponding to the sample with the fewest sequences. Subsequent analyses of alpha and beta diversity were performed using this normalized data.

Alpha diversity was calculated for analyzing complexity of biodiversity for a sample through the use of six indices: observed-species, Chao1 [22], Shannon [23], Simpson [24], ACE [25], and good-coverage. All these indices in our samples were calculated with QIIME (Version 1.7.0) [18] and displayed with R software (Version 2.15.3) [26]. Tukey’s test was used to determine the significance of differences in alpha diversity indices between groups. Beta diversity on both weighted and unweighted unifrac distances [27,28,29] were calculated using QIIME software (Version 1.7.0). Principal Coordinate Analysis (PCoA) was performed to display principal coordinates from complex, multidimensional data, which were visualized using R software (Version 2.15.3). MetaStats, a strict statistical method based on their abundance via multiple-hypothesis tests for sparsely sampled features and false discovery rate, was employed for differential-abundance analysis [30]. Level of statistical significance for all analyses was set at *p* ≤ 0.05.

## 3. Results

DENV infection was not detected in any of the larvae tested. The sequencing analysis of 24 libraries yielded a total of 3,079,063 effective reads (Appendix A). The OTU abundance (in terms of numbers of sequences of each) was evaluated in eight sample categories (based on species, developmental stage and gender of adults, habitat water: Table 1). In total, 7195 OTUs were identified (Appendix A). Most (5319) were bacteria spread across 39 phyla, 114 classes, 281 orders, 485 families, 992 genera, and 496 identified species (Appendix A). The other 1876 OTUs belonged to unknown taxa. The number of core OTUs, present in all categories, was 268 (Figure 2A and Appendix A). The largest numbers of category-specific OTUs were found in adult males of *Ae. aegypti* (743 OTUs) and of *Ae. albopictus* (489 OTUs). 

### 3.1. Community Structure, Richness, and Diversity of the Aedes Mosquito Microbiota

The microbiota community structure was examined by principal coordinate analysis (PCoA; Figure 2B). In the UniFrac analysis performed on the three different developmental stages of each *Aedes* mosquito species and their habitat water, the unweighted UniFrac PCoA shows very clear separation of clusters of the *Aedes* species from each other and from their larval habitat water. This pattern is less distinct according to the weighted UniFrac analysis, but the mosquito data were still clearly separated from the larval habitat water clusters.

Microbial alpha diversity, measured through Shannon and Simpson indices, and evenness were significantly greater in the abdominal microbiota of *Ae. aegypti* adults than in the *Ae. albopictus* adult categories (*p* < 0.05) (Figure 2C). Microbial richness was assessed using the ACE and Chao1 indices: there were no significant differences between larval and adult sample categories in either *Aedes* species. However, there was a significant difference in diversity and richness between mosquitos and habitat water categories (Appendix A).

### 3.2. Bacterial Composition among Aedes Species, Development Stages, and Their Habitat Water

The composition of the bacterial communities differed between mosquito species and developmental stages. At the phylum level, the bacterial communities of both mosquito species and their habitat water were dominated by Proteobacteria with abundances ranging from 37% in the *Ae. aegypti* larval habitat water category (AeyW) to 96% in the *Ae. albopictus* adult female category (AlbF) (Figure 3A). Firmicutes was a co-dominant phylum in all three categories of *Ae. aegypti*, whereas this was not the case for *Ae. albopictus*. Spirochaetota represented 10% to 15% of reads in larval and adult males of *Ae. albopictus*. Chloroflexi, Myxococcota, Nitrospirota, and Planctomycetota were almost absent from mosquitos (<1%) but were abundant in larval habitat waters (Figure 3A). At the family level, we observed more variation across sample groups (Figure 3B). The larval-stage categories of *Ae. aegypti* and *Ae. albopictus* were dominated by Spirochaetaceae, Planococcaceae, Yersiniaceae, and Comamonadaceae. The Comamonadaceae were found in high abundance in water (~9%) but in low abundance in adult mosquitos (<1%). Anaplasmataceae were the most prevalent bacterial family found in *Ae. albopictus* adults, whereas Enterobacteriaceae were abundant in *Ae. aegypti* adults. Markedly, Aeromonadaceae were abundant in females of *Ae. aegypti* and *Ae. albopictus* (Figure 3B).

The genus-level annotation is shown in Figure 3C. The clustering heatmap showed relative abundance of the 35 most-abundant genera in the bacterial communities of each sample category. The results provide deeper insight into differences in bacterial communities between Aedes species, development stages, and their larval habitat water. The genus *Aeromonas* was significantly more abundant in the female mosquitos (AeyF and AlbF categories) than in male mosquitos and larvae. The abundance of *Wolbachia* in both males and females of *Ae. albopictus* was significantly higher than in *Ae. aegypti*. In the AeyM category, the wide range of genera represented included the *Clostridium innocuum* group, *Coprobacillus*, *Blautia*, *Parasutterella*, *Akkermansia*, *Bifidobacterium*, *Castellaniella*, *Morganella*, *Klebsiella*, *Escherichia-Shigella*, and *Clostridioides*. Almost all those genera found in the AeyM group were also found in the AeyF group, although at a lower abundance (Figure 3C and Appendix A). The bacterial communities in larvae and their habitat water were dominated by different suites of genera, as shown in the genus-abundance heatmap (Figure 3C). 

### 3.3. Bacterial Species Differences between Ae. aegypti and Ae. albopictus

Taxa differing significantly between the two mosquito species were evaluated via Metastats based on relative abundance data. This revealed that the most common bacterial genera were significantly more abundant in *Ae. aegypti* than in *Ae. albopictus* (Figure 4A and Appendix A). Only the genus *Wolbachia* occurred at significantly higher frequencies in male *Ae. albopictus* (Figure 4A). Noticeably, the genus *Blautia* was significantly enriched in *Ae. aegypti*.

### 3.4. Changes in Microbiota between Larval and Adult Mosquitoes

The bacterial communities were significantly different between mosquito larval and adult stages (Figure 4B and Appendix A). The proportion of the genus *Aquabacterium* was significantly lower in the adult than in the larva, which was observed in both *Ae. aegypti* and *Ae. albopictus*. From larva to adult female of *Ae. aegypti*, four abundant genera differed significantly (*p* < 0.05). These included *Wolbachia*, *Blautia*, *Clostridioides,* and *Shinella,* all of which increased in abundance in the adult stage. Between the larval stage and adult male *Ae. aegypti*, abundance of the genera *Escherichia-Shigella*, *Clostridium innocuum* group, *Blautia*, and *Parasutterella* all increased. In *Ae. albopictus*, the microbial community of the larval stage was dominated by the genus *Serratia*, but this was replaced in abundance by the genus *Wolbachia* in AlbF and AlbM (Figure 4B).

## 4. Discussion

Although mosquito microbial communities have been reported previously using the 16S rRNA gene sequencing-based approach [31,32], there is little information available concerning the pattern of microbiota communities in the two important dengue vectors, *Ae. aegypti* and *Ae. Albopictus*, in northeastern Thailand [7,8,31,32]. The purpose of this study was to identify the microbiota of mosquito species collected in an endemic area of dengue fever to better understand the pattern of mosquito-associated microbial communities and their diversity. We sequenced the microbiomes of developmental stages of *Ae. aegypti* and *Ae. albopictus* from the field and from water at their breeding sites. All larvae were tested for DENV RNA, but none was detected in spite of reports of transovarial dengue-virus transmission of *Ae. aegypti* in Thailand [33,34].

The study found the bacterial community associated with *Ae. aegypti* and *Ae. albopictus* was mosquito-species specific in terms of bacterial abundance. The bacterial community changed according to developmental stage, from the larval stage to the adult stage, with increased microbial diversity in the adult stage (Figure 2C). The microbiome of *Ae. aegypti* was more abundant and diverse than that of *Ae. albopictus*, especially in the adult stage (Figure 2C and Figure 4A), even though both species of mosquitoes were sampled when newly eclosed and when they had not fed. The microbial communities in mosquitoes differed substantially from those in the water of their larval habitats (Figure 2B). According to unweighted UniFrac distances (providing a qualitative assessment of bacterial lineages in different communities), the samples were clustered mainly by larval habitat water and by *Aedes* species, suggesting that the main effect was whether lineages could survive in each of the different mosquito species. Weighted UniFrac distances (providing a quantitative measure of community differences that are due to changes in relative taxon abundance) showed that most of the mosquitoes were grouped together in loose clusters separated from the larval habitat water, indicating differences in microbial communities between environment and mosquitoes.

A small number of bacterial OTUs (3.7%, 268 OTUs) were shared across all sample groups in the study. These were almost all among the most abundant bacterial taxa overall, but their relative proportions varied by mosquito species and developmental stage. These common OTUs appeared to represent bacteria acquired by larvae from their aquatic habitat that were also able to persist across developmental stages until reaching and colonizing the adult abdomen (transstadial transmission). Similar transstadial transmission of some members of the larval bacterial community was previously demonstrated in *Ae. aegypti* [10].

Our data revealed a Proteobacteria-dominated microbiome. This is the most common phylum seen in mosquito genera such as *Aedes* and *Anopheles* [35,36,37]. Particularly, Proteobacteria accounted for more than 80% of bacteria in field-collected *Ae. albopictus* in one previous study in Thailand [38]. The phylum Firmicutes was also very abundant in the *Ae. aegypti* group in the present study.

The phyla of bacteria identified in the water of the mosquito larval habitats can vary, as noted in previous studies on other mosquito species [39,40,41]. The presence of only a few dominant bacterial taxa shared between mosquito larvae and water from their habitats appears to support the hypothesis that the mosquito provides a selective environment where only a few aquatic bacteria can survive. This characteristic has been observed in previous investigations on *Aedes* spp. [7,41,42].

Indeed, the most abundant bacterial taxa in this study are commonly found in *Aedes* mosquitoes [9,31,42]. At the family level, Comamonadaceae was abundant in larval habitat water that we sampled and members of this family were abundant only in larval stages of both mosquito species. Comamonadaceae is one of the most common families in freshwater environments and is also prevalent in mosquitoes [43,44]. For example, the genera *Aquabacterium* and *Hydrogenophaga* are members of this family. There are differences in abundance of these between *Ae. aegypti* and *Ae. albopictus* and between larval and adult stages (Figure 3 and Figure 4B). For example, the bacterial community in *Ae. aegypti* larvae was dominated by *Aquabacterium*, an aquatic bacterium, but this was largely replaced in adult females by the genus *Wolbachia* (family Anaplasmataceae) and by *Escherichia-Shigella* (family Enterobacteriaceae) in male mosquitoes (Figure 4B). *Wolbachia* was prominent in adult *Ae. albopictus* of both sexes, whereas larvae contained a large proportion of reads from the genus *Serratia* (family Yersiniaceae) (Figure 4B). The presence of certain bacteria such as *Serratia* and *Wolbachia* has been shown to give protection against pathogen infections in several mosquito species [45,46,47]. The higher abundance of *Serratia* in bacterial communities of larval stages, particularly in *Ae. albopictus*, contrasts with previous findings of this genus being abundant in adult mosquito reproductive organs [31,37,48,49] and midgut [38]. 

*Serratia* spp. are known to inhibit infection of *Anopheles* by *Plasmodium* [46] and also was found in field and laboratory-reared *Ae. aegypti* cultivable midgut microbiota [50]. The polypeptides of gut-inhabiting *Serratia odorifera* can increase the DENV-2 susceptibility of *Ae. aegypti* females by blocking the prohibitin molecule present on the surface of the midgut of these females [51]. *Serratia marcescens* can facilitate arboviral infection and enhance viral dissemination via a secreted protein that digests membrane-bound mucins on the mosquito gut epithelium [52] that are associated with immune response of the mosquito gut [53]. Moreover, *Serratia oryzae* may also promote the development of insecticide resistance [54]. However, the role of the microbiota in mosquito immune response and reproduction requires much further research.

The taxon *Wolbachia* is a promising biocontrol agent [55]. This genus of Gram-negative endosymbiotic bacteria includes obligate, intracellular maternally inherited organisms that occur in many insect species in cells of the reproductive tissues and non-reproductive tissue such as midgut and salivary glands [12,56,57]. In *Ae. aegypti*, *Wolbachia* can inhibit viral replication, dissemination, and transmission in experimental studies, while in *Ae. albopictus* it has no effect on the replication of dengue virus but can reduce the viral load in mosquito salivary glands and limit transmission [12,56,57]. In this study, *Wolbachia* was found in larvae and adult males and females of both *Aedes* species from natural sources but was much more abundant in *Ae. albopictus*. This difference between the mosquito species (Appendix A) might explain why *Ae. aegypti* is the main vector transmitting dengue in Southeast Asia, while *Ae. albopictus* acts as a secondary vector [56]. Native *Ae. aegypti* larvae have also been found to host *Wolbachia* in Malaysia [12], a country neighboring Thailand. Here, we have demonstrated for the first time the presence of *Wolbachia* in *Ae. aegypti* larvae and female adults collected in Thailand. This new finding is surprising, as to our knowledge *Ae. aegypti* adult mosquitoes have never been reported to host *Wolbachia*. Further investigation is required to characterize the strain type of *Wolbachia* found in *Ae. aegypti* larvae and adults, especially in adult female *Ae. aegypti*. Native *Wolbachia* infection in *Ae. aegypti* may render virus-control strategies involving artificial *Wolbachia* introduction redundant.

Microbial communities differed between the two sexes of mosquitoes. The genus *Aeromonas* (family Aeromonadaceae) was observed in higher abundance in females of both species than in male mosquitos. This genus is commonly found in the guts of mosquitos [58]. Previous reports have demonstrated that *Ae. aegypti* fed with *Aeromonas* sp. were more susceptible to DENV-2, although the underlying mechanism was not revealed [59]. *Aeromonas hydrophila*, a potent chitinolytic bacterium, was the most abundant *Aeromonas* species in female *Aedes* in our study (Figure 3C and Appendix A). This species is commonly detected in *Anopheles arabiensis* mosquitos from South Africa [35], and it was discovered in female *Ae. aegypti* cultured in the laboratory but not in wild mosquitoes in Brazil [60]. Chitinase enzymes produced by *A. hydrophila* (SBK1 strain) have promise as effective mosquitocidal agents [61]. *Lysinibacillus sphaericus*, another possible biocontrol agent for mosquito larvae [62], was found in the highest abundance in *Ae. aegypti* larvae in the present study (Appendix A). However, it is not clear what effects these bacteria might have on the mosquitoes we investigated. Further work is needed to find bacterial species that can be applied for control of mosquitoes and mosquito-borne diseases [63].

Here, we assessed larval and adult mosquitoes of separate sexes and water from their larval habitat, provided information on the biology of mosquitoes, and provided an initial profile of their microbial communities in an endemic area of dengue in northeastern Thailand. Although many factors influence the bacterial community in mosquitoes, the most abundant taxa can always be detected, even in different geographies or environments. The finding also demonstrates the likelihood of transstadial transmission of bacterial communities between larval and adult mosquitoes. Future research should focus on more in-depth analyses of each mosquito-associated microbe.

## Figures and Tables

**Figure 1 insects-14-00184-f001:**
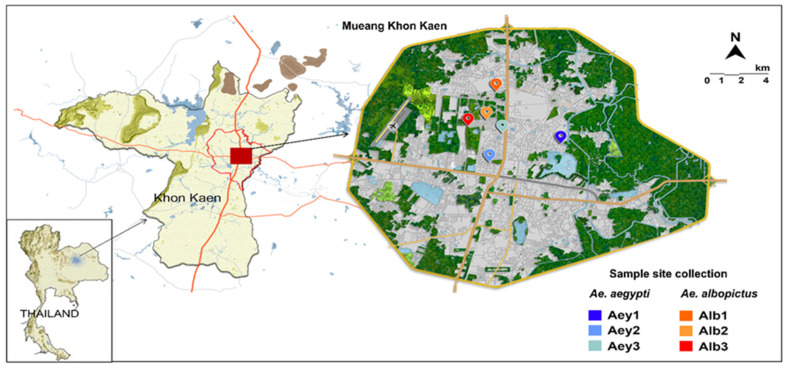
Map of the larvae collection site.

**Figure 2 insects-14-00184-f002:**
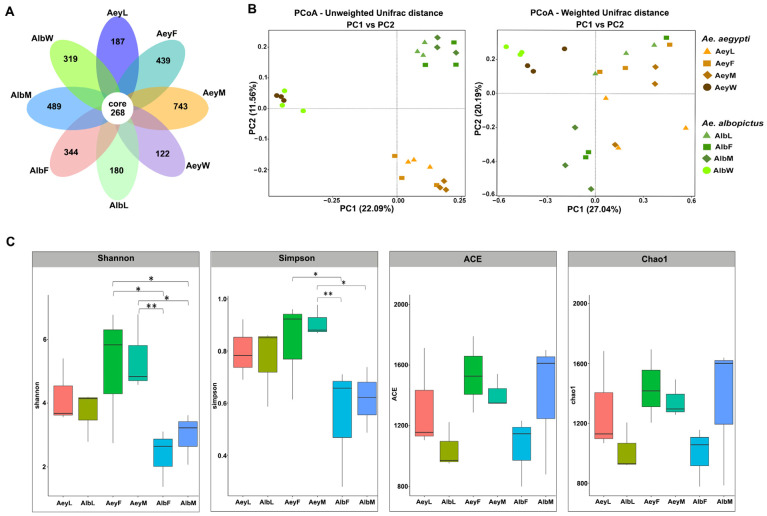
Community structure and diversity of *Aedes* mosquito microbiota. (**A**) The flower diagram shows the number of core OTUs and the number of OTUs unique to each individual category. (**B**) Principal coordinate analysis (PCoA) of the *Aedes* species and larval habitat water microbiota in all samples. PCoA plot based on unweighted UniFrac distance matrices and weighted UniFrac distance matrices. UniFrac distances were calculated for all OTUs. Each data point represents values from one sample with color/shape identifying individual samples. (**C**) Boxplot of alpha diversity indices. Shannon and Simpson indices reflect the OTU diversity in samples. The ACE and Chao1 indices estimate the OTU richness in samples. Boxes represent the interquartile range, lines indicate medians, and whiskers indicate the range. Wilcoxon and Tukey’s tests were used to detect statistically significant differences between categories (*, *p* ≤ 0.05; **, *p* ≤ 0.01).

**Figure 3 insects-14-00184-f003:**
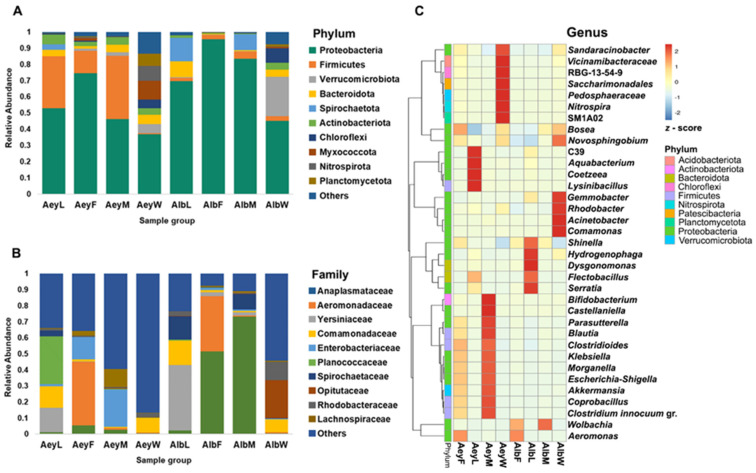
Composition of microbial community of *Ae. aegypti* and *Ae. albopictus* and larval habitat water. The top 10 taxa in terms of relative abundance at (**A**) Phylum level and (**B**) Family level. (**C**) Taxonomic abundance cluster heatmap showing the relative abundance of the top 35 genera for each sample category plotted by the absolute z-score. Samples are clustered according to the similarity between their constituents and arranged in horizontal order. The colors in the heatmap refer to the species abundance, according to the color bar on the right. Red and blue colors show the high and low abundance of a genus, respectively.

**Figure 4 insects-14-00184-f004:**
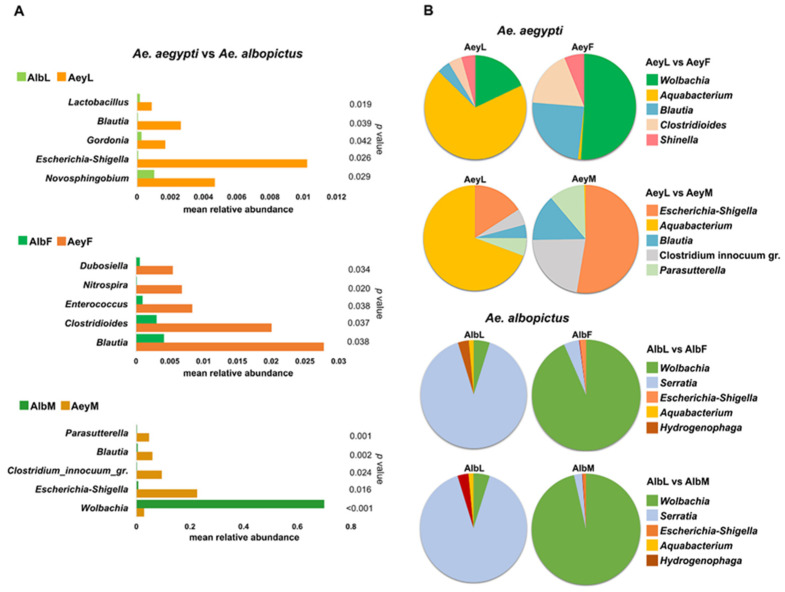
The top 5 genera in terms of significant differences in abundance between and within mosquito species. (**A**) Comparisons of developmental stages/genders between *Ae. aegypti* and *Ae. albopictus*. (**B**) Pairwise comparisons between larval and adult stages of each *Aedes* species. The significance of observed abundance differences among groups was evaluated using MetaStat. Statistically significant differences between groups were set at *p* < 0.05.

**Table 1 insects-14-00184-t001:** Number of samples used for construction of DNA libraries (*n* = 24).

Sample Type	*Ae*. *aegypti* (Aey)		*Ae*. *albopictus* (Alb)	
	Library Category Name	Number of Libraries *	Library Category Name	Number of Libraries *
Larva (L)	AeyL	3	AlbL	3
Male (M)	AeyM	3	AlbM	3
Female (F)	AeyF	3	AlbF	3
Larval habitat water (W)	AeyW	3	AlbW	3

* Pools of 10 larvae, or adult males, or adult females per library.

## Data Availability

The authors confirm that the data supporting the findings of this study are available within the article and its Appendix A. All sequence reads have been deposited at the NCBI Sequence Read Archive (SRA) under project accession number PRJNA919511.

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
