# Peer review of "Microbiome Composition and Microbial Community Structure in Mosquito Vectors Aedes aegypti and Aedes albopictus in Northeastern Thailand, a Dengue-Endemic Area"

_insects, 2023, doi:10.3390/insects14020184_

Round 1

Reviewer 1 Report

This manuscript examines the microbial community in Aedes aegypti and Aedes albopictus collected in Thailand. Species and larval habitats are treated separately and analyzed by a variety of methods for significant differences in composition. This is an interesting study that adds to the knowledge base. Recommend minor revisions. The observation of Wolbachia in Aedes aegypti needs to be addressed more thoroughly.

The manuscript is generally well written. The methods are mostly well described although a few improvements could be made as noted below. 

Lines 147-152: What program was used for determination mean abundance, relative abundance and normalization? It is unclear as written.

Lines 194-195: The difference between the unweighted and weighted results should be briefly addressed in the discussion.

Lines 201-202: I do not see a figure that shows the water.

Lines 224-225: Wolbachia are not considered to be found in Aedes aegypti and this underlies an entire strategy of using injected Wolbachia as a control method. This manuscript shows significant presence in Ae. albopictus as expected but surprisingly shows a distinct presence, although lower, in Ae. aegypti. This must be well addressed in the discussion.

Line 244: Suggest replacing "commonest" with "most common."

Lines 259-260: Wolbachia increased in adult female Ae. aegypti? Must be addressed in discussion.

Line 315: Please replace "form" with "from."

Lines 332-342: This portion of the discussion relating to the presence of Wolbachia should definitely be expanded. As noted, Wolbachia is a promising biocontrol agent, particularly for Aedes aegypti where it is considered to be natively absent (DOI: 10.1016/j.cell.2009.11.042). Patent infection with Wolbachia must be initially introduced since it is not naturally occurring (doi: 10.4269/ajtmh.15-0608). This manuscript shows that it does occur natively. This should be more thoroughly addressed. First, has the presence of native Wolbachia been seen in Aedes aegypti previously? If not, this is an important find. Second, regardless of whether it has or has not been found previously, discussing the potential implications of native Wolbachia infection on the control strategy of artificial introduction seems critical.

Author Response

Reviewer 1

Comments and Suggestions for Authors

This manuscript examines the microbial community in Aedes aegypti and Aedes albopictus collected in Thailand. Species and larval habitats are treated separately and analyzed by a variety of methods for significant differences in composition. This is an interesting study that adds to the knowledge base. Recommend minor revisions. The observation of Wolbachia in Aedes aegypti needs to be addressed more thoroughly.

The manuscript is generally well written. The methods are mostly well described although a few improvements could be made as noted below.

Lines 147-152: What program was used for determination mean abundance, relative abundance and normalization? It is unclear as written.

Reply: We have added in lines 150-155 “The OTUs in the individual samples were divided among eight categories (species, life-stage and gender, and habitat water), and the mean abundances of each OTU were computed using QIIME for further analysis. The relative abundance of OTUs was normalized using a sequence number standard corresponding to the sample with the fewest sequences. Subsequent analyses of alpha and beta diversity were performed using this normalized data.”

Lines 194-195: The difference between the unweighted and weighted results should be briefly addressed in the discussion.

Reply: We have now dealt with this in the discussion section. Please see lines 288-296.

Lines 201-202: I do not see a figure that shows the water.

Reply: We have added this as Figure S1.

Lines 224-225: Wolbachia are not considered to be found in Aedes aegypti and this underlies an entire strategy of using injected Wolbachia as a control method. This manuscript shows significant presence in Ae. albopictus as expected but surprisingly shows a distinct presence, although lower, in Ae. aegypti. This must be well addressed in the discussion.

Reply: Native Aedes aegypti larvae were also found to contain Wolbachia in Malaysia, a country neighboring Thailand (12. Teo, C.H.J.; Lim, P.K.C.; Voon, K.; Mak, J.W. Detection of dengue viruses and Wolbachia in Aedes aegypti and Aedes albopictus larvae from four urban localities in Kuala Lumpur, Malaysia. Trop Biomed 2017, 34, 583–597.). We have added new sentences in discussion section “Native Ae. aegypti larvae have also been found to host Wolbachia in Malaysia [12], a country neighboring Thailand. Here, we have demonstrated for the first time the presence of Wolbachia in Ae. aegypti larvae and female adults collected in Thailand. This new finding is surprising, as to our knowledge Ae. aegypti adult mosquitoes have never been reported to host Wolbachia. Further investigation is required to characterize the strain type of Wolbachia found in Ae. aegypti larvae and adults, especially in adult female Ae. aegypti. Native Wolbachia infection in Ae. aegypti may render virus-control strategies involving artificial Wolbachia introduction redundant.” Please see lines 354-362.

Line 244: Suggest replacing "commonest" with "most common."

Reply: we have done this. Please see line 248.

Lines 259-260: Wolbachia increased in adult female Ae. aegypti? Must be addressed in discussion.

Reply: We have added discussion of this (see above)

Line 315: Please replace "form" with "from."

Reply: we have corrected this. Please see line 327.

Lines 332-342: This portion of the discussion relating to the presence of Wolbachia should definitely be expanded. As noted, Wolbachia is a promising biocontrol agent, particularly for Aedes aegypti where it is considered to be natively absent (DOI: 10.1016/j.cell.2009.11.042). Patent infection with Wolbachia must be initially introduced since it is not naturally occurring (doi: 10.4269/ajtmh.15-0608). This manuscript shows that it does occur natively. This should be more thoroughly addressed. First, has the presence of native Wolbachia been seen in Aedes aegypti previously? If not, this is an important find. Second, regardless of whether it has or has not been found previously, discussing the potential implications of native Wolbachia infection on the control strategy of artificial introduction seems critical.

Reply: we added the sentence “Native Ae. aegypti larvae have also been found to host Wolbachia in Malaysia [12], a country neighboring Thailand. Here, we have demonstrated for the first time the presence of Wolbachia in Ae. aegypti larvae and female adults collected in Thailand. This new finding is surprising, as to our knowledge Ae. aegypti adult mosquitoes have never been reported to host Wolbachia. Further investigation is required to characterize the strain type of Wolbachia found in Ae. aegypti larvae and adults, especially in adult female Ae. aegypti. Native Wolbachia infection in Ae. aegypti may render virus-control strategies involving artificial Wolbachia introduction redundant.” Please see lines 354-362. 

Finally, we would like to thank the reviewer very much for the kind suggestions. Your comments are supportive and helpful. 

Reviewer 2 Report

In the manuscript entitled “Microbiome composition and microbial community structure in mosquito vectors Aedes aegypti and Aedes albopictus in northeastern Thailand, a dengue-endemic area”, Rodpai et al. collected Aedes aegypti and albopictus mosquito larvae to quantify their microbiome. They also let some of these larvae evolve into adults to analyze the evolution of the microbiome between these two stages of development and between males and females.

They found significant differences between all possible comparisons such as the disappearance of Serratia in adults or the greater presence of Wolbachia in albopictus and more particularly in males.

These differences are observed but not explained, they are unfortunately not related to any sensitivity to the Dengue virus since none of the samples were positive for this virus.

The analysis methods used seem to me to be adequate and correctly set out in the methods chapter.

Major comment :

I don't understand why no effort seems to have been made to collect aegypti and albopictus from the same sites. The authors should explain this point.

The fact of having chosen 3 sites for albo and 3 sites for aegy necessarily leads to the appearance of a bias linked to the sites for the comparison between albo and aegy.

I understand that each library is only linked to one collection site, but this is not clearly specified. If this is the case, the absence of bias related to the sampling site must clearly be underlined in the results chapter of the PCoA. If this is not the case, the possibility of a bias must clearly be evoked.

Minor comment :

As all the samples are negative for DENV, I do not see the need to mention DENV in the title and in the abstract of this article.

Author Response

Reviewer 2

Comments and Suggestions for Authors

In the manuscript entitled “Microbiome composition and microbial community structure in mosquito vectors Aedes aegypti and Aedes albopictus in northeastern Thailand, a dengue-endemic area”, Rodpai et al. collected Aedes aegypti and albopictus mosquito larvae to quantify their microbiome. They also let some of these larvae evolve into adults to analyze the evolution of the microbiome between these two stages of development and between males and females.

They found significant differences between all possible comparisons such as the disappearance of Serratia in adults or the greater presence of Wolbachia in albopictus and more particularly in males.

These differences are observed but not explained, they are unfortunately not related to any sensitivity to the Dengue virus since none of the samples were positive for this virus.

The analysis methods used seem to me to be adequate and correctly set out in the methods chapter.

Reply: The manuscript has been improved as suggested. Please see below.

Major comment :

I don't understand why no effort seems to have been made to collect aegypti and albopictus from the same sites. The authors should explain this point.

The fact of having chosen 3 sites for albo and 3 sites for aegy necessarily leads to the appearance of a bias linked to the sites for the comparison between albo and aegy.

Reply: Larvae of the two species do not generally co-occur in our study area. For clarity, we have added the sentence “Sites where Ae. aegypti was collected were close to homes, whereas Ae. albopictus was generally found in forests or patches of shrubby vegetation: larvae of the two species did not co-occur.” Please see revised manuscript, lines 75-77.

I understand that each library is only linked to one collection site, but this is not clearly specified. If this is the case, the absence of bias related to the sampling site must clearly be underlined in the results chapter of the PCoA. If this is not the case, the possibility of a bias must clearly be evoked.

Reply: No bias was linked to the sampling sites because larvae of the two Aedes species do not occur together in the same habitat. Please see the response to a previous query. 

Minor comment :

As all the samples are negative for DENV, I do not see the need to mention DENV in the title and in the abstract of this article.

Reply: We used the title “Microbiome composition and microbial community structure in mosquito vectors Aedes aegypti and Aedes albopictus in north-eastern Thailand, a dengue-endemic area” to highlight the fact that the work was done in a dengue-endemic area. This alerts readers in this field when comparisons are made with studies in nonendemic area of dengue. Thus, we wish to retain this title.

Finally, we greatly appreciate the kind suggestions made by the reviewer. Your comments are supportive and helpful.